# Superconducting metamaterials for waveguide quantum electrodynamics

Mohammad Mirhosseini[1,2,3], Eunjong Kim[1,2,3], Vinicius S. Ferreira[1,2,3], Mahmoud Kalaee[1,2,3], Alp Sipahigil [1,2,3], Andrew J. Keller[1,2,3] & Oskar Painter[1,2,3]

Embedding tunable quantum emitters in a photonic bandgap structure enables control of dissipative and dispersive interactions between emitters and their photonic bath. Operation in the transmission band, outside the gap, allows for studying waveguide quantum electrodynamics in the slow-light regime. Alternatively, tuning the emitter into the bandgap results in finite-range emitter–emitter interactions via bound photonic states. Here, we couple a transmon qubit to a superconducting metamaterial with a deep sub-wavelength lattice constant ($\lambda/60$). The metamaterial is formed by periodically loading a transmission line with compact, low-loss, low-disorder lumped-element microwave resonators. Tuning the qubit frequency in the vicinity of a band-edge with a group index of $n_g = 450$, we observe an anomalous Lamb shift of $-28$ MHz accompanied by a 24-fold enhancement in the qubit lifetime. In addition, we demonstrate selective enhancement and inhibition of spontaneous emission of different transmon transitions, which provide simultaneous access to short-lived radiatively damped and long-lived metastable qubit states.

[1] Kavli Nanoscience Institute, California Institute of Technology, Pasadena, CA 91125, USA. [2] Thomas J. Watson, Sr., Laboratory of Applied Physics, California Institute of Technology, Pasadena, CA 91125, USA. [3] Institute for Quantum Information and Matter, California Institute of Technology, Pasadena, CA 91125, USA. Correspondence and requests for materials should be addressed to O.P. (email: opainter@caltech.edu)

Cavity quantum electrodynamics (QED) studies the interaction of an atom with a single electromagnetic mode of a high-finesse cavity with a discrete spectrum[1,2]. In this canonical setting, a large photon–atom coupling is achieved by repeated interaction of the atom with a single photon bouncing many times between the cavity mirrors. Recently, there has been much interest in achieving strong light–matter interaction in a cavity-free system such as a waveguide[3,4]. Waveguide QED refers to a system where a chain of atoms are coupled to a common optical channel with a continuum of electromagnetic modes over a large bandwidth. Slow-light photonic crystal waveguides are of particular interest in waveguide QED because the reduced group velocity near a bandgap preferentially amplifies the desired radiation of the atoms into the waveguide modes[5–7]. Moreover, in this configuration an interesting paradigm can be achieved by placing the resonance frequency of the atom inside the bandgap of the waveguide[8–11]. In this case, the atom cannot radiate into the waveguide but the evanescent field surrounding it gives rise to a photonic bound state[9]. The interaction of such localized bound states has been proposed for realizing tunable spin–exchange interaction between atoms in a chain[12,13], and also for realizing effective non-local interactions between photons[14,15].

While achieving efficient waveguide coupling in the optical regime requires the challenging task of interfacing atoms or atomic-like systems with nanoscale dielectric structures[16–20], superconducting circuits provide an entirely different platform for studying the physics of light–matter interaction in the microwave regime[4,21]. Development of the field of circuit QED has enabled fabrication of tunable qubits with long coherence times and fast qubit gate times[22,23]. Moreover, strong coupling is readily achieved in coplanar platforms due to the deep sub-wavelength transverse confinement of photons attainable in microwave waveguides and the large electric dipole of superconducting qubits[24]. Microwave waveguides with strong dispersion, even "bandgaps" in frequency, can also be simply realized by periodically modulating the geometry of a coplanar transmission line[25]. Such an approach was recently demonstrated in a pioneering experiment by Liu and Houck[26], whereby a qubit was coupled to the localized photonic state within the bandgap of a modulated coplanar waveguide (CPW). Satisfying the Bragg condition in a periodically modulated waveguide requires a lattice constant on the order of the wavelength, however, which translates to a device size of approximately a few centimeters for complete confinement of the evanescent fields in the frequency range suitable for microwave qubits. Such a restriction significantly limits the scaling in this approach, both in qubit number and qubit connectivity.

An alternative approach for tailoring dispersion in the microwave domain is to take advantage of the metamaterial concept. Metamaterials are composite structures with sub-wavelength components, which are designed to provide an effective electromagnetic response[27,28]. Since the early microwave work, the electromagnetic metamaterial concept has been expanded and extensively studied across a broad range of classical optical sciences[29–31]; however, their role in quantum optics has remained relatively unexplored, at least in part due to the lossy nature of many sub-wavelength components. Improvements in design and fabrication of low-loss superconducting circuit components in circuit QED offer a new prospect for utilizing microwave metamaterials in quantum applications[32]. Indeed, high quality-factor superconducting components such as resonators can be readily fabricated on a chip[33], and such elements have been used as a tool for achieving phase-matching in near quantum-limited traveling wave amplifiers[34,35] and for tailoring qubit interactions in a multi-mode cavity QED architecture[36].

In this paper, we utilize an array of coupled lumped-element microwave resonators to form a compact bandgap waveguide with a deep sub-wavelength lattice constant ($\lambda/60$) based on the metamaterial concept. In addition to a compact footprint, these sort of structures can exhibit highly nonlinear band dispersion surrounding the bandgap, leading to exceptionally strong confinement of localized intra-gap photon states. We present the design and fabrication of such a metamaterial waveguide, and characterize the resulting waveguide dispersion and bandgap properties via interaction with a tunable superconducting transmon qubit. We measure the Lamb shift and lifetime of the qubit in the bandgap and its vicinity, demonstrating the anomalous Lamb shift of the fundamental qubit transition as well as selective inhibition and enhancement of spontaneous emission for the first two excited states of the transmon qubit.

## Results

**Band-structure analysis and spectroscopy.** We begin by considering the circuit model of a CPW that is periodically loaded with microwave resonators as shown in the inset to Fig. 1a. The Lagrangian for this system can be constructed as a function of the node fluxes of the resonator and waveguide sections $\Phi_n^b$ and $\Phi_n^a$[37]. Assuming periodic boundary conditions and applying the rotating wave approximation, we derive the Hamiltonian for this system and solve for the energies $\hbar\omega_{\pm,k}$ along with the corresponding eigenstates $|\pm,k\rangle = \hat{\alpha}_{\pm,k}|0\rangle$ as (see Supplementary Note 1)

$$\omega_{\pm,k} = \frac{1}{2}\left[(\Omega_k + \omega_0) \pm \sqrt{(\Omega_k - \omega_0)^2 + 4g_k^2}\right], \qquad (1)$$

$$\hat{\alpha}_{\pm,k} = \frac{(\omega_{\pm,k} - \omega_0)}{\sqrt{(\omega_{\pm,k} - \omega_0)^2 + g_k^2}}\hat{a}_k + \frac{g_k}{\sqrt{(\omega_{\pm,k} - \omega_0)^2 + g_k^2}}\hat{b}_k. \qquad (2)$$

Here, $\hat{a}_k$ and $\hat{b}_k$ describe the momentum-space annihilation operators for the bare waveguide and bare resonator sections, the index $k$ denotes the wavevector, and the parameters $\Omega_k$, $\omega_0$, and $g_k$ quantify the frequency of traveling modes of the bare waveguide, the resonance frequency of the microwave resonators, and coupling rate between resonator and waveguide modes, respectively. The operators $\hat{\alpha}_{\pm,k}$ represent quasi-particle solutions of the composite waveguide, where far from the bandgap the quasi-particle is primarily composed of the bare waveguide mode, while in the vicinity of $\omega_0$ most of its energy is confined in the microwave resonators.

Figure 1a depicts the numerically calculated energy bands $\omega_{\pm,k}$ as a function of the wavevector $k$. It is evident that the dispersion has the form of an avoided crossing between the energy bands of the bare waveguide and the uncoupled resonators. For small gap sizes, the midgap frequency is close to the resonance frequency of uncoupled resonators $\omega_0$, and unlike the case of a periodically modulated waveguide, there is no fundamental relation tying the midgap frequency to the lattice constant in this case. The form of the band structure near the higher cut-off frequency $\omega_{c+}$ can be approximated as a quadratic function $(\omega - \omega_{c+}) \propto k^2$, whereas the band structure near the lower band-edge $\omega_{c-}$ is inversely proportional to the square of the wavenumber $(\omega - \omega_{c-}) \propto 1/k^2$. The analysis above has been presented for resonators which are capacitively coupled to a waveguide in a parallel geometry; a similar band structure can also be achieved using series inductive coupling of resonators (see Supplementary Note 1 and Supplementary Fig. 1).

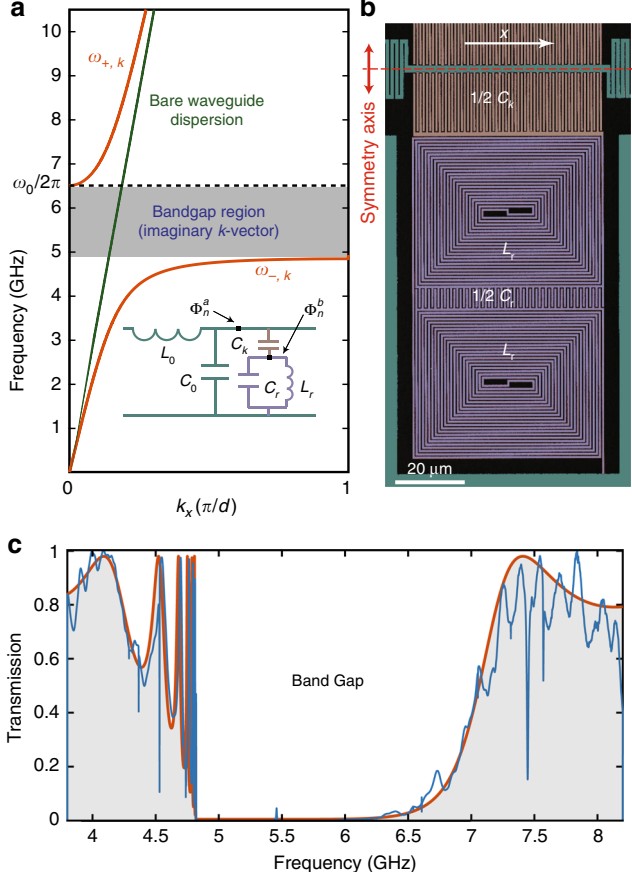

**Fig. 1** Microwave metamaterial waveguide. **a** Dispersion relation of a CPW loaded with a periodic array of microwave resonators (red curve). The green line shows the dispersion relation of the waveguide without the resonators. Inset: circuit diagram for a unit cell of the periodic structure. **b** Scanning electron microscope (SEM) image of a fabricated capacitively coupled microwave resonator, here with a wire width of 500 nm. The resonator region is false-colored in purple, the waveguide central conductor and the ground plane are colored green, and the coupling capacitor is shown in orange. We have used pairs of identical resonators symmetrically placed on the two sides of the transmission line to preserve the symmetry of the structure. **c** Transmission measurement for the realized metamaterial waveguide made from 9 unit cells of resonator pairs with a wire width of 1μm, repeated with a lattice constant of $d = 350$ μm. The blue curve depicts the experimental data and the red curve shows the lumped-element model fit to the data

**Physical realization using lumped-element resonators**. A coplanar microwave resonator is often realized by terminating a short segment of a coplanar transmission line with a length set to an integer multiple of $\lambda/4$, where $\lambda$ is the wavelength corresponding to the fundamental resonance frequency[25,33]. However, it is possible to significantly reduce the footprint of a resonator by using components that mimic the behavior of lumped elements. We have used the design presented in ref. [38] to realize resonators in the frequency range of 6–10 GHz. This design provides compact resonators by placing interdigital capacitors at the anti-nodes of the charge waves and double spiral coils near the peak of the current waves at the fundamental frequency (see Fig. 1b). The symmetry of this geometry results in the suppression of the second harmonic frequency and thus the elimination of an undesired bandgap at twice the fundamental resonance frequency of the band-gap waveguide. A more subtle design criterion is that

the resonators be of high impedance. Use of high impedance resonators allows for a larger photonic bandgap and greater waveguide–qubit coupling. For the waveguide QED application of interest this enables denser qubit circuits, both spatially and spectrally.

The impedance of the resonators scales roughly as the inverse square-root of the pitch of the wires in the spiral coils. Complicating matters is that smaller wire widths have been found to introduce larger resonator frequency disorder due to kinetic inductance effects[39]. Here, we have selected an aggressive resonator wire width of 1μm and fabricated a periodic array of $N = 9$ resonator pairs coupled to a CPW with a lattice constant of $d = 350$. The resonators are arranged in identical pairs placed on opposite sides of the central waveguide conductor to preserve the symmetry of the waveguide. In addition, the center conductor of each CPW section is meandered over a length of 210 μm so as to increase the overall inductance of the waveguide section which also increases the bandgap. Further details of the design criteria and lumped element parameters are given in Supplementary Note 2. The fabrication of the waveguide is performed using electron-beam deposited Al film (see Methods). Figure 1c shows the measured power transmission through such a finite-length metamaterial waveguide. Here 50-Ω CPW segments, galvanically coupled to the metamaterial waveguide, are used at the input and output ports. We find a midgap frequency of 5.83 GHz and a bandgap extent of 1.82 GHz for the structure. Using the simulated value of effective refractive index of 2.54, the midgap frequency gives a lattice constant-to-wavelength ratio of $d/\lambda \approx 1/60$.

**Disorder and Anderson localization**. Fluctuations in the electromagnetic properties of the metamaterial waveguide along its length, such as the aforementioned resonator disorder, results in random scattering of traveling waves. Such random scattering can lead to an exponential extinction of propagating photons in the presence of weak disorder and complete trapping of photons for strong disorder, a phenomenon known as the Anderson localization of light[40]. Similarly, absorption loss in the resonators results in attenuation of wave propagation which adds a dissipative component to the effective localization of fields in the metamaterial waveguide. Figure 2a shows numerical simulations of the effective localization length as a function of frequency when considering separately the effects of resonator frequency disorder and loss (see Supplementary Note 3 for details of independent resonator measurements used to determine frequency variation (0.5%) and loss parameters (intrinsic Q-factor of $7.2 \times 10^4$) for this model). In addition to the desired localization of photons within the bandgap, we see that the effects of disorder and loss also limit the localization length outside the bandgap. In the lower transmission band where the group index is largest, the localization length is seen to rapidly approach zero near the band-edge, predominantly due to disorder. In the upper transmission band where the group index is smaller, the localization length maintains a large value of $6 \times 10^3$ periods all the way to the band-edge. Within the bandgap the simulations show that the localization length is negligibly modified by the levels of loss and disorder expected in the resonators of this work, and is well approximated by the periodic loading of the waveguide alone which can be simply related to the inverse of the curvature of the transmission bands of a loss-less, disorder-free structure[13]. These results indicate that, even with practical limitations on disorder and loss in such metamaterial waveguides, a range of photon length scales of nearly four orders of magnitude should be accessible for frequencies within a few hundred MHz of the band-edges of the gap (see Supplementary Note 4).

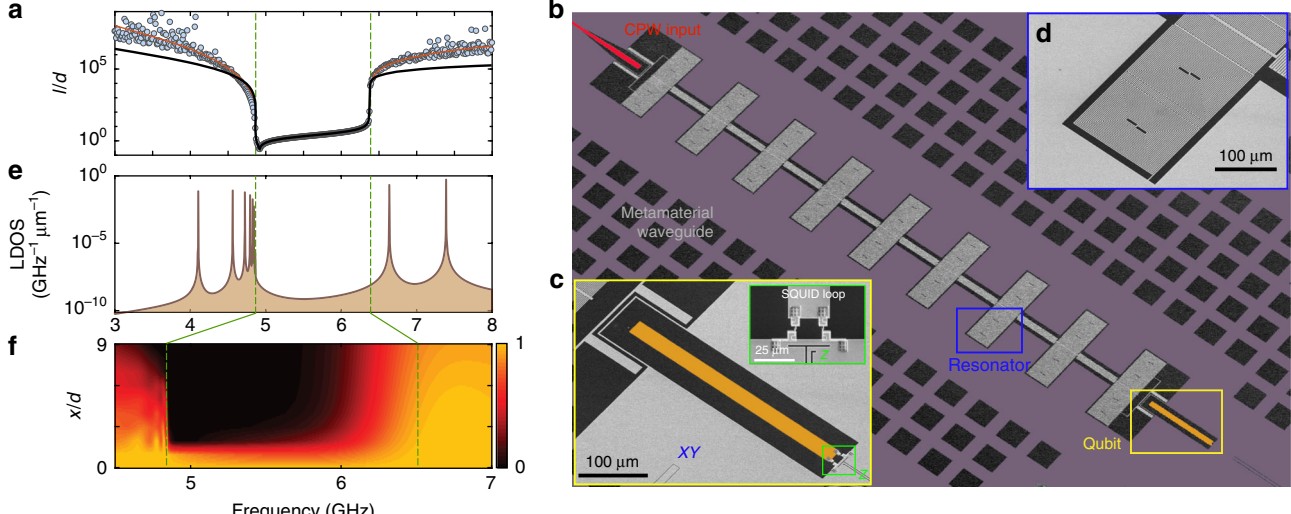

**Fig. 2** Disorder effects and qubit–waveguide coupling. **a** Calculated localization length for a loss-less metamaterial waveguide with structural disorder (blue circles). The nominal waveguide parameters are determined from the fit to a lumped element model (including resonator loss) to the transmission data in Fig. 1. Numerical simulation has been performed for $N = 100$ unit cells, averaged over $10^5$ randomly realized values of the resonance frequency $\omega_0$, with the standard deviation $\delta\omega_0/\omega_0 = 0.5\%$. The vertical green lines represent the extent of the bandgap region. The red curve outside the gap is an analytic model based on ref. [53]. For comparison, the solid black curve shows the calculated effective localization length without resonator frequency disorder but including resonator loss. **b** SEM image of the fabricated qubit–waveguide system. The metamaterial waveguide (gray) consists of 9 periods of the resonator unit cell. The waveguide is capacitively coupled to an external CPW (red) for reflective read-out. **c** The transmon qubit is capacitively coupled to the resonator at the end of the array. The Z drive is used to tune the qubit resonance frequency by controlling the external flux bias in the superconducting quantum interference device (SQUID) loop. The XY drive is used to coherently excite the qubit. **d** Capacitively coupled microwave resonator. **e** Calculated local density of states (LDOS) at the qubit position for a metamaterial waveguide with a length of 9 unit cells and open boundary conditions (experimental measurements of LDOS tabulated in Supplementary Table 1). The band-edges for the corresponding infinite structure are marked with vertical green lines. **f** Normalized electromagnetic energy distribution along the waveguide vs. qubit frequency for the coupled qubit–waveguide system. The vertical axis marks the distance from the qubit ($x/d$) in units of the lattice period $d$

**Anomalous Lamb shift near the band-edge**. To further probe the electromagnetic properties of the metamaterial waveguide we couple it to a superconducting qubit. In this work, we use a transmon qubit[22] with the fundamental resonance frequency $\omega_{ge}/2\pi = 7.9$ GHz and Josephson energy to single electron charging energy ratio of $E_J/E_C \approx 100$ at zero flux bias (details of our qubit fabrication methods can also be found in ref. [41]). Figure 2b shows the geometry of the device where the qubit is capacitively coupled to one end of the waveguide and the other end is capacitively coupled to a 50-$\Omega$ CPW transmission line. This geometry allows for forming narrow individual modes in the transmission band of the metamaterial, which can be used for dispersive qubit state read-out[42] from reflection measurements at the 50-$\Omega$ CPW input port (see Supplementary Note 2 and Supplementary Table 1). Figure 2e, f shows the theoretical photonic LDOS and spatial photon energy localization versus frequency for this finite length qubit–waveguide system. Within the bandgap the qubit is self-dressed by virtual photons which are emitted and re-absorbed due to the lack of escape channels for the radiation. Near the band-edges surrounding the bandgap, where the LDOS is rapidly varying with frequency, this results in a large anomalous Lamb shift of the dressed qubit frequency[10,43]. Figure 3a shows the measured qubit transition frequency shift from the expected bare qubit tuning curve as a function of frequency. Shown for comparison is the circuit theory model frequency shift of a finite structure with $N = 9$ periods (blue solid curve) alongside that of an infinite length waveguide (red dashed curve). It is evident that the qubit frequency is repelled from the band-edges on the two sides of the bandgap, a result of the strongly asymmetric density of states in these two regions. The measured frequency shift at the lower frequency band-edge is 43 MHz, in good agreement with the circuit theory model. Note that at the lower frequency band-edge where

the localization length approaches zero due to the anomalous dispersion (see Fig. 2a), boundary-effects in the finite structure do not significantly alter the Lamb shift. Near the upper-frequency band-edge, where finite-structure effects are non-negligible due to the weaker dispersion and corresponding finite localization length, we measure a qubit frequency shift as large as −28 MHz. This again is in good correspondence with the finite structure model; the upper band-edge of the infinite length waveguide occurs at a slightly lower frequency with a slightly smaller Lamb shift.

**Enhancement and suppression of spontaneous emission**. Another signature of the qubit–waveguide interaction is the change in the rate of spontaneous emission of the qubit. Tuning the qubit into the bandgap changes the localization length of the waveguide photonic state that dresses the qubit (see Fig. 2f). Since the finite waveguide is connected to an external port which acts as a dissipative environment, the change in localization length $\ell(\omega)$ is accompanied by a change in the lifetime of the qubit $T_{rad}(\omega) \propto e^{2x/\ell(\omega)}$, where $x$ is the total length of the waveguide (see Supplementary Note 5). In addition to radiative decay into the output channel, losses in the resonators in the waveguide also contribute to the qubit's excited state decay. Using a low power probe in the single-photon regime we have measured intrinsic Q-factors of $7.2 \pm 0.4 \times 10^4$ for the individual waveguide resonances between 4.6 and 7.4 GHz. Figure 3b shows the measured qubit lifetime ($T_1$) as a function of its frequency in the bandgap. The solid blue curve in Fig. 3b shows a fitted theoretical curve which takes into account the loss in the waveguide along with a phenomenological intrinsic lifetime of the qubit ($T_{1,i} = 10.8$ μs). The dashed red curve shows the expected qubit lifetime for an infinite waveguide length. Qualitatively, the measured lifetime of

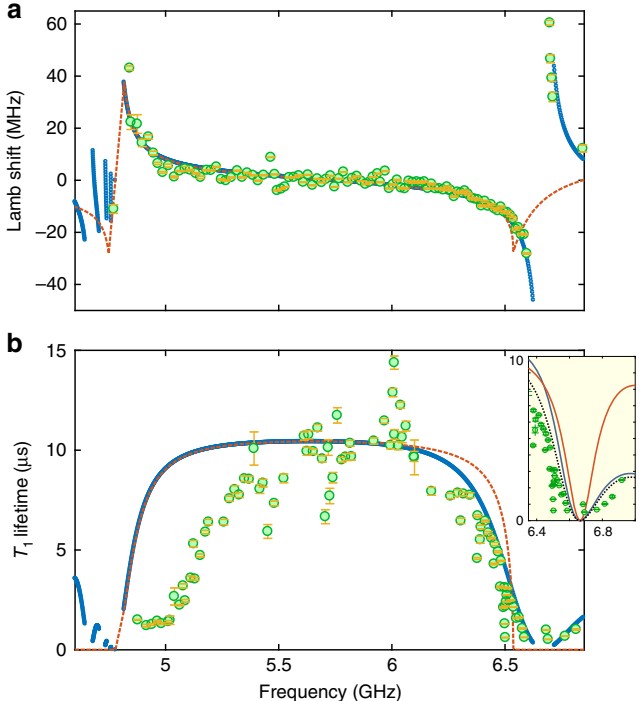

**Fig. 3** Measured dispersive and dissipative qubit dynamics. **a** Lamb shift of the qubit transition vs. qubit frequency. **b** Lifetime of the excited qubit state vs. qubit frequency. Open circles show measured data. The solid blue line (dashed red line) is a theoretical curve from the circuit model of a finite (infinite) waveguide structure. For determining the Lamb shift from measurement, the bare qubit frequency is calculated as a function of flux bias $\Phi$ as $\hbar\omega_{ge} = \sqrt{8E_C E_J(\Phi)} - E_C$ using the extracted values of $E_C$, $E_J$, and assuming the symmetric SQUID flux bias relation $E_J(\Phi) = E_{J,\,max}\cos(2\pi\Phi/\Phi_0)$[22]. The lifetime characterization is performed in the time domain where the qubit is initially excited with a $\pi$ pulse through the XY drive. The excited state population, determined from the state-dependent dispersive shift of a close-by band-edge waveguide mode, is measured subsequent to a delay time during which the qubit freely decays. Inset to (**b**) shows a zoomed in region of the qubit lifetime near the upper band-edge. Solid blue (red) lines show the circuit model contributions to output port radiation (structural waveguide loss), adjusted to include a frequency independent intrinsic qubit lifetime of 10.86 μs. The black dashed line shows the cumulative theoretical lifetime

the qubit behaves as expected; the qubit lifetime drastically increases inside the bandgap region and is reduced in the transmission bands. More subtle features of the measured lifetime include multiple, narrow Fano-like spectral features deep within the bandgap. These features arise from what are believed to be interference between parasitic on-chip modes and low-$Q$ modes of our external copper box chip packaging. In addition, while the measured lifetime near the upper band-edge is in excellent agreement with the finite waveguide theoretical model, the data near the lower band-edge shows significant deviation. We attribute this discrepancy to the presence of low-$Q$ parasitic resonances, observable in transmission measurements between the qubit XY drive line and the 50-$\Omega$ CPW port. Possible candidates for such spurious modes include the asymmetric "slotline" modes of the waveguide, which are weakly coupled to our symmetrically grounded CPW line but may couple to the qubit. Further study of the spectrum of these modes and possible methods for suppressing them will be a topic of future studies.

Focusing on the upper band-edge, we plot as an inset to Fig. 3b a zoom-in of the measured qubit lifetime along with theoretical

estimates of the different components of qubit decay. Here, the qubit decay results from two dominant effects: detuning-dependent coupling to the lossy resonances in the transmission band of the waveguide, and emission into the output port of the finite waveguide structure. The former effect is an incoherent phenomenon arising from a multi-mode cavity-QED picture, whereas the latter effect arises from the coherent interference of band-edge resonances which can be related to the photon bound state picture and resulting localization length. Owing to the weaker dispersion at the upper band-edge, the extent of the photon bound state has an appreciable impact on the qubit lifetime in the $N = 9$ finite length waveguide. This is most telling in the strongly asymmetric qubit lifetime around the first waveguide resonance in the upper transmission band. Quantitatively, the slope of the radiative component of the lifetime curve in the bandgap near the band-edge can be shown to be proportional to the group delay (see Supplementary Note 6), $|\partial T_{rad}/\partial\omega| = T_{rad}\tau_{delay}$. The corresponding group index, $n_g \equiv \tau_{delay}/x$, is a property of the waveguide independent of its length $x$. Here, we measure a slope corresponding to a group index $n_g \approx 450$, in good correspondence with the circuit model of the lossy metamaterial waveguide.

The sharp variation in the photonic LDOS near the metamaterial waveguide band-edges may also be used to engineer the multi-level dynamics of the qubit. A transmon qubit, by construct, is a nonlinear quantum oscillator and thus has a multilevel energy spectrum. In particular, a third energy level ($|f\rangle$) exists at the frequency $\omega_{gf} = 2\omega_{ge} - E_C/\hbar$. Although the transition g–f is forbidden by selection rules, the f–e transition has a dipole moment that is $\sqrt{2}$ larger than the fundamental transition[22]. This is consistent with the scaling of transition amplitudes in a harmonic oscillator and results in a second transition lifetime that is half of the fundamental transition lifetime for a uniform density of states in the electromagnetic environment of the oscillator. The sharply varying density of states in the metamaterial, on the other hand, can lead to strong suppression or enhancement of the spontaneous emission for each transition. Figure 4 shows the measured lifetimes of the two transitions for two different spectral configurations. In the first scenario, we enhance the ratio of the lifetimes $T_{eg}/T_{fe}$ by situating the fundamental transition frequency deep inside in the bandgap while having the second transition positioned near the lower transmission band. The situation is reversed in the second configuration, where the fundamental frequency is tuned to be near the upper frequency band while the second transition lies deep inside the gap. In our fabricated qubit, the second transition is about 300 MHz lower than the fundamental transition frequency at zero flux bias, which allows for achieving large lifetime contrast in both configurations.

## Discussion

Looking forward, we anticipate that further refinement in the engineering and fabrication of the devices presented here should enable metamaterial waveguides approaching a lattice constant-to-wavelength ratio of $\lambda/1000$, with limited disorder and a bandgap-to-midgap ratio in excess of 50% (see Supplementary Note 7). Such compact, low loss, low disorder superconducting metamaterials can help realize more scalable superconducting quantum circuits with higher levels of complexity and functionality in several regards. They offer a method for densely packing qubits—both in spatial and frequency dimensions—with isolation from the environment and controllable connectivity achieved via bound qubit–waveguide polaritons[7,13,44]. Moreover, the ability to selectively modify the transition lifetimes provides simultaneous access to long-lived metastable

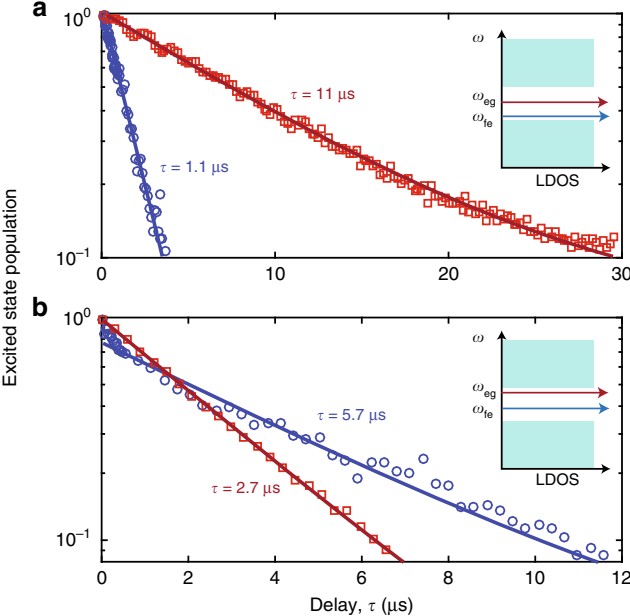

**Fig. 4** State-selective enhancement and inhibition of radiative decay. **a** Measurement with the e–g transition tuned deep into the bandgap ($\omega_{eg}/2\pi = 5.37$GHz), with the f–e transition near the lower transmission band ($\omega_{fe}/2\pi = 5.01$ GHz). **b** Measurement with the e–g transition tuned near the upper transmission band ($\omega_{eg}/2\pi = 6.51$ GHz), with the f–e transition deep in the bandgap ($\omega_{fe}/2\pi = 6.17$ GHz). For measuring the f–e lifetime, we initially excite the third energy level |f⟩ via a two-photon $\pi$ pulse at the frequency of $\omega_{gf}/2$. Following the population decay in a selected time interval, the population in |f⟩ is mapped to the ground state using a second $\pi$ pulse. Finally, the ground state population is read using the dispersive shift of a close-by band-edge resonance of the waveguide. g–e (f–e) transition data shown as red squares (blue circles)

qubit states as well as short-lived states strongly coupled to waveguide modes. This approach realizes a transmon qubit system with state-dependent bound state localization lengths, which can be used as a quantum nonlinear media for propagating microwave photons[15,45,46], or as recently demonstrated, to realize spin-photon entanglement and high-bandwidth itinerant single microwave photon detection[47,48]. Combined, these attributes provide a unique platform for studying the many-body physics of quantum photonic matter[49–52].

## Methods

**Device fabrication**. The devices used in this work are fabricated on silicon substrates [Float zone (FZ) grown, 500 thickness, >10 kOhm-cm resistivity]. The ground plane, metamaterial waveguide, and qubit capacitor are patterned by electron-beam lithography followed by electron-beam evaporation of 120 nm Al at a rate of 1 nm/s. A liftoff process is performed in n-methyl-2-pyrrolidone at 80 °C for 1.5 h. The Josephson junctions are fabricated using double-angle electron beam evaporation of suspended bridges, following similar techniques as in ref. [41].

**Device characterization**. The fabricated devices are characterized in a dilution refrigerator with a base temperature of $T_f \approx 7$ mK. The input coaxial lines are thermalized at each stage of the fridge with a series of attenuators to reduce the Johnson thermal noise from the room-temperature environment. The output signal is directed through a pair of isolators at the mixing-chamber stage of the fridge and is subsequently sent into an amplifier chain consisting of a HEMT amplifier (Low Noise Factory LNF-LNC4_8C) the 4-K fridge stage and a low-noise amplifier (Miteq AFS42-00101200-22-10P-42) at room temperature. Frequency-domain characterization is performed using a two-port vector network analyzer (VNA). The transmission ($S_{21}$) and reflection ($S_{11}$, separated by a circulator) signals are selectively directed to the output line by means of a mechanical RF switch. For time-domain characterization, a pair of pulse sequences are synthesized for exciting the qubit and for performing read-out. A Tektronix AWG5014C arbitrary

waveform generator (AWG) is used to generate I–Q signals at the IF frequency (<200 MHz), and the signals are upconverted in a pair of mixers with local oscillator tones supplied by CW microwave sources (Rohde & Schwarz SMB100A). The output read-out signal is downconverted with a mixer and is registered using a 1 GS/s PCIe digitizer (AlazarTech ATS9870).

## Data availability

The data that support the findings of this study are available from the corresponding author (O.P.) upon reasonable request.

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

## Acknowledgements

We would like to thank Paul Dieterle, Ana Asenjo Garcia, and Darrick Chang for fruitful discussions regarding waveguide QED. This work was supported by the AFOSR MURI Quantum Photonic Matter (grant 16RT0696), the AFOSR MURI Wiring Quantum Networks with Mechanical Transducers (grant FA9550-15-1-0015), the Institute for Quantum Information and Matter, an NSF Physics Frontiers Center (grant PHY-1125565) with support of the Gordon and Betty Moore Foundation, and the Kavli Nanoscience Institute at Caltech. M.M. (A.J.K., A.S.) gratefully acknowledges support from a KNI (IQIM) Postdoctoral Fellowship.

## Author contributions

M.M., V.S.F. and O.P. came up with the concept. M.M., A.S. and O.P. planned the experiment. M.M., M.H., V.S.F., A.J.K. and E.K. performed the device design and fabrication. M.M., E.K. and A.S. performed the measurements. M.M., E.K., A.S. and O.P. analyzed the data. All authors contributed to the writing of the manuscript.

## Additional information

**Competing interests:** The authors declare no competing interests.

