## [Peer Review File · Nature Communications]

REVIEWERS' COMMENTS:

Reviewer #1 (Remarks to the Author):

As I explained in my previous report, this work is of high quality and should be of interest to a specialized audience such as that of Nature Communications. I have read the replies of the authors to both my report and the reports of the other two reviewers. I find that the authors have responded very well and in detail to all the questions raised in the reports. The manuscript has been updated in places, which has improved the paper. As I write below, the authors should in some cases consider including more details from their replies in the supplementary material. After the authors have considered my remarks below, I recommend that the manuscript is published in Nature Communications.

1. In response to my comment 6 (about Eq. 2 in the main text and Eq. 25 in the supplementary material), the text in the supplementary material has been updated in a satisfactory way, but I believe that the text in connection with Eq. 2 can still be improved to be clearer.
2. The content of the reply to my comment 8 (regarding the basis for some estimates given in the conclusion) should be made available to all readers, not just the referees. I think that the authors should include it somewhere in the supplementary material. I also think that the authors could include much of their reply to comment 6 of reviewer 3 in the same place, making this a section with details about the prospects for scaling up the experimental system under consideration.
3. The changes in reply to my comment 9 (about the rotating-wave approximation) are mostly OK, but a period is missing at the end of the 2nd sentence after Eq. 20 and that last inequality should be " $C_k \ll C_0$ or $C_k \ll C_r$ ", not " $C_k \ll C_0, C_r$ ", right? At least that is how I understand Eq. 13.

Reviewer #3 (Remarks to the Author):

I found the responses given by the authors very competent and to-the-point. The manuscript is in excellent shape and ready for publication. The question to be addressed is whether the degree of novelty of the paper is sufficient to warrant publication in a high-impact journal. As pointed out by all three referees, while the architecture presented looks promising for the field, the physics of an atom closed to the band edge has already been explored by the Houck group, and sub-wavelength components are not new in microwave engineering with superconductors. For a journal like Nat Commun, publishing "important advances of significance to specialists within each field", this is a tough call. Given the increasing attention that these structures are receiving in the field, and the foreseen follow-up experiments and applications, I would eventually lean towards publication.

Author Response:

The authors thank all the reviewers for their careful and detailed review of our manuscript. Please find below the authors' responses (**marked in red**) to each of the reviewers' points (**in black**)

Reviewer #1 (Remarks to the Author):

As I explained in my previous report, this work is of high quality and should be of interest to a specialized audience such as that of Nature Communications. I have read the replies of the authors to both my report and the reports of the other two reviewers. I find that the authors have responded very well and in detail to all the questions raised in the reports. The manuscript has been updated in places, which has improved the paper. As I write below, the authors should in some cases consider including more details from their replies in the supplementary material. After the authors have considered my remarks below, I recommend that the manuscript is published in Nature Communications.

We thank the reviewer for the favorable assessment of our manuscript. Below, we have listed our response to each of the reviewer's comments.

1. In response to my comment 6 (about Eq. 2 in the main text and Eq. 25 in the supplementary material), the text in the supplementary material has been updated in a satisfactory way, but I believe that the text in connection with Eq. 2 can still be improved to be clearer.

We have added a comment regarding the definition of eigenstates above Eq. 2 in the main text, in a similar fashion to the comment in the supplementary note.

2. The content of the reply to my comment 8 (regarding the basis for some estimates given in the conclusion) should be made available to all readers, not just the referees. I think that the authors should include it somewhere in the supplementary material. I also think that the authors could include much of their reply to comment 6 of reviewer 3 in the same place, making this a section with details about the prospects for scaling up the experimental system under consideration.

Following the reviewer's suggestion, we have included the discussions on the technical challenges of scaling in a supplementary note ("Supplementary Note 7"), and referenced it in the main text.

3. The changes in reply to my comment 9 (about the rotating-wave approximation) are mostly OK, but a period is missing at the end of the 2nd sentence after Eq. 20 and that last inequality should be " $C_k \ll C_0$ or $C_k \ll C_r$ ", not " $C_k \ll C_0, C_r$ ", right? At least that is how I understand Eq. 13.

We thank the reviewer for pointing out the typo. We have modified the sentence to remove any ambiguity in interpretation.

Reviewer #3 (Remarks to the Author):

I found the responses given by the authors very competent and to-the-point. The manuscript is in excellent shape and ready for publication. The question to be addressed is whether the degree of novelty of the paper is sufficient to warrant publication in a high-impact journal. As pointed out by all three referees, while the architecture presented looks promising for the field, the physics of an atom closed to the band edge has already been explored by the Houck group, and sub-wavelength components are not new in microwave engineering with superconductors. For a journal like Nat Commun, publishing "important advances of significance to specialists within each field", this is a tough call.

Given the increasing attention that these structures are receiving in the field, and the foreseen follow-up experiments and applications, I would eventually lean towards publication.

We thank the reviewer for the favorable assessment of our manuscript.